# Work Task Association with Lead Urine and Blood Concentrations in Informal Electronic Waste Recyclers in Thailand and Chile

**DOI:** 10.3390/ijerph182010580

**Published:** 2021-10-09

**Authors:** Abas Shkembi, Kowit Nambunmee, Siripond Jindaphong, Denisse Parra-Giordano, Karla Yohannessen, Pablo Ruiz-Rudolph, Richard L. Neitzel, Aubrey Arain

**Affiliations:** 1Department of Environmental Health Sciences, University of Michigan, Ann Arbor, MI 48109, USA; ashkembi@umich.edu (A.S.); rneitzel@umich.edu (R.L.N.); 2Urban Safety Innovation Research Group (USIR), School of Health Science, Mae Fah Luang University, Chiang Rai 57100, Thailand; kowit.nam@mfu.ac.th (K.N.); siripond.jin13@lamduan.mfu.ac.th (S.J.); 3Departamento de Enfermería, Facultad de Medicina, Universidad de Chile, Santiago 8380453, Chile; drparra@uchile.cl; 4Programa de Salud Ambiental, Instituto de Salud Poblacional, Facultad de Medicina, Universidad de Chile, Santiago 8380453, Chile; k.yohannessen@gmail.com (K.Y.); pabloruizr@uchile.cl (P.R.-R.); 5Department of Environmental Health and Safety, Oakland University, Rochester, MI 48309, USA

**Keywords:** e-waste, lead, metals, occupational health, boosted regression trees

## Abstract

The informal recycling of electronic waste (“e-waste”) is a lucrative business for workers in low- and middle-income countries across the globe. Workers dismantle e-waste to recover valuable materials that can be sold for income. However, workers expose themselves and the surrounding environment to hazardous agents during the process, including toxic metals like lead (Pb). To assess which tools, tasks, and job characteristics result in higher concentrations of urine and blood lead levels among workers, ten random samples of 2 min video clips were analyzed per participant from video recordings of workers at e-waste recycling sites in Thailand and Chile to enumerate potential predictors of lead burden. Blood and urine samples were collected from participants to measure lead concentration. Boosted regression trees (BRTs) were run to determine the relative importance of video-derived work variables and demographics, and their relationship with the urine and blood concentrations. Of 45 variables considered, five job characteristics consisting of close-toed shoes (relative importance of 43.9%), the use of blunt striking instruments (14%), bending the back (5.7%), dismantling random parts (4.4%), and bending the neck (3.5%) were observed to be the most important predictors of urinary Pb levels. A further five job characteristics, including lifting objects <20 lbs. (6.2%), the use of screwdrivers (4.2%), the use of pliers/scissors (4.2%), repetitive arm motion (3.3%), and lifting objects >20 pounds (3.2%) were observed to be among the most important factors of blood Pb levels. Overall, our findings indicate ten job characteristics that may strongly influence Pb levels in e-waste recycling workers’ urine and blood.

## 1. Introduction

The informal recycling of e-waste, or waste containing electrical or electronic components, is an increasingly essential source of income among the world’s lower socioeconomic populations [1,2]. According to the United Nations, 53.6 million metric tons of e-waste was generated worldwide in 2019 [3]. Globally, e-waste often flows from high-income countries to low- and middle-income countries or low-income communities within richer countries [4]. In these settings, recycling offers a valuable source of income for workers with low socioeconomic status due to the recovery of useful materials and components [5].

In most countries, the informal sector dominates recycling activity [2,6]. This informal sector lacks government regulations and controls imposed on the formal sector to dismantle and dispose of electronic waste [2] safely; they also lack access to health services if workers experience occupational health impacts [7,8]. In the informal sector, the recovery of valuable metals from e-waste is usually prioritized, often using rudimentary methods that adversely impact health and the environment [2]. Furthermore, due to the informality, workers may have little or no training in the safe extraction of valuable metals from e-waste, and often perform these tasks without any personal protective equipment (PPE) or other safety devices [4,9]. As a result, e-waste decommissioning can include hazardous activities such as breaking glass cathode-ray tubes (CRTs), burning plastics, and the acid leaching of waste to extract valuable metals [10]. These activities can endanger the health of recyclers, their families, and the surrounding environment, and can lead to elevated levels of heavy metals in the body that enter through inhalation or ingestion once released into the environment [2,11,12]. 

Despite the potential to cause health problems from occupational and environmental exposure to the hazards of e-waste, the economic benefit associated with recycling e-waste may outweigh the perceived health risks of workers. Because of this, researchers have proposed to study conditions to move to more sustainable informal e-waste processing [13], including the study of conditions and activities that lead to increased exposures [12,13]. For instance, in the past, greater exposures were observed for activities such as dismantling and the use of blunt instruments and electric saws [14,15], or the open-air combustion of materials [4], while other tasks such as e-waste repair or refurbishment may have lower exposures. Given the potential effects of different work practices and activities on heavy metal exposure (including different contexts), information on these factors can help account for metal urine and blood levels in e-waste recycling workers [2]. In this sense, it is key to identify actions that reduce these exposures so that e-waste recycling workers can maintain their income through recycling while simultaneously protecting their health and that of their communities.

The objective of this study was to evaluate the association of demographic characteristics, work tasks, use of tools, personal behaviors, and other work aspects with the levels of lead (Pb) in urine and blood among informal recycling workers of e-waste. We studied recycling populations in Thailand and Chile, locations where the context of e-waste differs, and subsequently provides variability in the e-waste activities under study. We have previously described some aspects of our research in the study communities in both countries [16,17,18], and this analysis was conducted on a subset of participants from both countries. Thailand is a middle-income country with a Gini wealth equality index of roughly 35 out of 100 [19], and a known receptor of e-waste from developed countries [13]. There is minimal regulation on e-waste recycling activities, and there is a general lack of knowledge about e-waste, databases or inventories, and the absence of environmental management practices, while e-waste imports have been accumulating rapidly [20]. As a consequence, in the summer of 2020, the Thai government prohibited the entry of e-waste from other countries [21]. In contrast, Chile is a highly urbanized and centralized high-income country [22] with greater wealth inequality than Thailand (Gini index of roughly 44) [23]. Chile has not been systematically identified as a receptor of e-waste [13], although it is one of the major producers of e-waste in Latin America [24]. There is a small, formal recycling industry in Chile, but the extent of informal recycling is unclear. In 2016, Chile established e-waste recycling laws primarily targeted toward formal recyclers and electronics manufacturers [24]. Thus, Thailand and Chile present two different “shades” of e-waste recycling: Thailand as a large importer of e-waste and a resulting, quickly expanding informal recycling sector, and Chile as a large producer of e-waste shared among both a formal and informal recycling sector.

## 2. Materials and Methods

### 2.1. Study Population and Design

To study e-waste recycling workers exposure and health effects, a comprehensive health and exposure assessment was conducted cross-sectionally in 2017, both in Thailand and Chile. In Thailand, four villages from a rural, low-income community in the Northeastern region, where informal e-waste recycling is prevalent, was selected; in Chile, informal workers were selected from two communities: the capital city of Santiago, and at the smaller town of Temuco. Additionally, workers from a formal plant in Chillán were included to increase variability in the e-waste recycling activities included in the sample. In each country, researchers from the University of Michigan partnered with researchers and students from local universities, local health care workers, government officials, and community members. 

Participants in each research site were recruited with the help of in-country partners. Because the focus of the study was informal e-waste recycling, no registry of workers exists, making random sampling techniques infeasible. Therefore, convenience sampling methods were used. The participants were e-waste recycling workers who were at least 18 years of age. An in-depth description of the e-waste sites that participated in this study, and characterization of the Chilean and Thai participants, were published previously [16,17]. Research methods were approved by the University of Michigan Institutional Review Board (HUM0014562) in the United States, Mae Fah Luang University in Thailand (REH-59104), and the University of Chile-Santiago (Archive Project N^o^ 101-2017; Act N^o^ 45). Informed consent was obtained from all participants before study enrollment.

### 2.2. Survey 

A 143-item survey instrument was used to gather data on the study participants. Interviews were conducted face-to-face with a native language speaker, and the responses were entered directly into a Qualtrics electronic survey form (Qualtrics, Seattle, WA, USA). Information on demographic characteristics, work history, e-waste tasks and tools, and health status was collected. 

### 2.3. Urine and Blood Pb Sampling

Blood samples were collected from participants by a locally registered nurse in each country. All sampling analysis was completed by the ISO-certified Thai Ministry of Public Health Central Laboratory in Bangkok, Thailand. Whole blood was analyzed for Pb, and urine was analyzed for Pb in addition to creatinine, which was used for the adjustment of urine Pb concentration. The methods for collecting and analyzing samples are detailed in our previously published works [17,25]. The results for blood samples are reported in µg/dL, and urine concentrations are reported in µg/g creatinine.

### 2.4. Video Data

Videos were collected of e-waste workers performing routine work tasks during work shifts in Thailand and Chile by positioning a GoPro camera (GoPro, San Mateo, CA, USA) at a fixed location within each worksite and oriented towards the work area (Figure 1). 

Videos were screened and edited to remove video segments where workers were not working or were not on screen. A randomly selected subset of 40 participants was selected for video data analysis. Among this subset, videos were sampled at regular intervals (with interval lengths determined based on the duration of the screened and edited footage) to yield five 2 min increments for a total of 10 min of footage per participant. A quantitative video assessment tool was then created to allow research staff to record the frequency of tasks, tool use, posturing, personal protective equipment, and other job characteristics observed in each video segment. These factors were enumerated across the entire duration of each 2 min segment into four categories: 0—never occurred or took zero seconds (s); 1—occurred one to three times, or up to 20 s; 2—occurred four to ten times or 21—80 s; and 3—occurred >10 times or more than 80 s. Before the formal evaluation of the video segments, a kappa test was run using a pilot set of videos viewed by five research assistants. This pilot evaluation yielded a kappa statistic of inter-rater agreement >0.6, which was determined to be acceptable. Research assistants then applied the quantitative tool to the entire video segment collection to create a numeric data set summarizing the observed job characteristics. The frequencies of each of these characteristics were averaged for each participant recorded in the videos, and that value was then assigned to represent that participant’s “typical” job characteristics. 

### 2.5. Statistical Analysis

SPSS 25 (SPSS Inc, Chicago, IL, USA) and R 4.0.2 (R Core Team) were used for all analyses. Concentrations of blood and urine were adjusted for each participant as described below before statistical analysis. Values below the analytical limit of detection (LOD) for each sample were adjusted by applying the LOD divided by the √2 [26]. Log transformations were then applied to the values where necessary to achieve normal distributions. All analyzes were stratified by country. Counts and frequencies summarized categorical variables, and continuous variables were summarized by means, standard deviations, medians, and maximums. Independent *t*-tests between the two populations were used to compare continuous variables, with equal variances not assumed. χ^2^ tests of homogeneity between the two populations were used to assess differences among the categorical variables. Both t-tests and χ^2^ tests were two-tailed and checked for significance at the following levels: *p* < 0.05, *p* < 0.01, and *p* < 0.001.

To determine which job characteristics were predictive of Pb levels in the blood and urine samples, boosted regression trees (BRTs) were utilized using the R package “gbm”. BRTs are a machine learning algorithm combining the techniques of boosting and regression trees together [27]. BRTs are used extensively in the field of ecology due to their ability to handle complex interactions and nonlinearity in data [28]. Due to the same complex and analytical issues often found in epidemiology and toxicology, there has been a push for utilizing BRTs in these fields as well [29,30], particularly due to their highly accurate and predictive capabilities [31].

Regression trees use recursive binary splits in a top-down, “greedy” approach. Each split creates two partitions which minimize the relative sum of squared errors from the split. This is top-down as it begins at the top of the tree and each split creates two new branches further down the tree. It is greedy since for each recursive binary split, the best split is made at each particular split without looking ahead to what would create a better tree.

As compared to fitting a single decision tree, boosting approaches learn slowly. BRTs grow many trees using information from previously grown trees in a sequential manner to minimize the residuals at each tree split. Each grown tree is allowed the same number of splits, known as the tree complexity. To introduce randomness into the model fit, a “bagging” factor of 0.8 is set. This means that the first tree is fit onto the dependent variable using 80 percent of randomly selected data from the training data. The second tree fits the residuals of the first tree using the bagging factor once more, and the model is updated. Each sequential tree repeats this method using the previous model until it reaches the pre-determined number of trees.

In this study, we ran BRTs using the demographic and observed job characteristics for each e-waste worker from the video data as predictors for the two outcomes of Pb concentrations in urine and blood. Despite having a small sample size (n = 40 workers) and many predictors (*p* = 46), BRTs can handle this combination, which is why they were preferred over more traditional regression techniques (e.g., multiple linear regression) for this analysis, which would yield unreliable results. Predictors that were highly correlated (r > 0.7) with other predictors were dropped from the BRT models. Predictors with no variability were also dropped. BRT models were fit onto 500 trees using a gaussian loss function with a learning rate of 0.1 and tree complexity of 10. During the BRT modeling process, the number of times a predictor is selected in the splitting step can be weighted by the squared improvement that the predictor provided to the model, averaged over the 500 trees, and scaled to a sum of 100 percent across all predictors. This is referred to as variable importance. A higher variable importance demonstrates a stronger relationship with the dependent variable than other predictors. Predictors with relative importance less than the importance expected by randomness were excluded from interpretation. This threshold is determined by 100/p, where p is the number of predictors in the model. Partial dependence plots of the predictors with relative importance above 100/p were also constructed to aid in the interpretability of the model results, and were fitted with a spline.

## 3. Results

### 3.1. Demographics

One hundred thirty e-waste recycling workers from Thailand, and 95 from Chile, participated in this study (N = 225 workers). The demographic characteristics for the sample are shown in Table 1. The mean in-community residence time among the Thai workers (41.7 ± 15.6 years) was significantly higher than the residence time of the Chilean workers (14.4 ± 15.1 years) (*p* < 0.05). The mean body mass index (BMI) of the Thai participants (24.9 ± 4.2 kg/m^2^) was significantly lower than that of the Chilean workers (30.1 ± 45.4 kg/m^2^) (*p* < 0.05). Among the categorical variables, the frequency of males among the 130 Thai workers was significantly lower than that of the 93 Chilean workers (χ^2^ = 7.1, df = 1, *p* < 0.01). There was a significant relationship between the marital status and the country of the e-waste workers (χ^2^ = 21.1, df = 5, *p* < 0.001), with 88.5% of Thai workers married, compared to 66.7% of Chilean workers. The frequency of workers employed by a business (as opposed to self-employed or working in a family business) is significantly higher among Chilean workers than the Thai workers (χ^2^ = 24.7, df = 1, *p* < 0.001). Lastly, the frequency of workers whose income was above the relevant national minimum wage was significantly higher among Chilean workers than Thai workers (χ^2^ = 12.3, df = 1, *p* < 0.001).

One hundred fifteen Thai and 82 Chilean workers were sampled for blood Pb, and 105 Thai and 86 Chilean workers for urinary Pb, as shown in Table 2. For both blood Pb levels and log levels of blood Pb, Thai workers were observed to have significantly higher concentrations than the Chilean workers (original—1.62 µg/dL, 95% CI: 1.09, 2.15; log—0.68 µg/dL, 95% CI: 0.50, 0.85). Similarly, Thai workers were observed to have significantly higher Pb levels in their urine than the Chilean workers (original—6.04 µg/g creatinine, 95% CI: 4.65, 7.44; log—1.09 µg/g creatinine, 95% CI: 0.91, 1.27). According to the US Occupational Safety and Health Administration (OSHA), blood lead levels (BLL) in a worker are considered elevated when it exceeds 5 µg/dL. Among our participants, 12% (23 participants) had elevated BLLs at the time of sampling, with 19 of those 23 (83%) participants from Thailand. No workers exceeded BLLs which would require medical removal from work (>50 µg/dL), with the maximum BLL recorded at 12.4 µg/dL.

### 3.2. Video Participants

Nineteen Thai workers and 21 Chilean workers were recorded for video analysis. Table 2 shows the levels of Pb in urine and blood among this subset of workers. Thai workers were observed to have significantly higher log levels of Pb in their blood when compared to the Chilean workers (1.05 µg/dL; 95% CI: 0.01, 0.77). With regards to urine, for both original and log levels of Pb, Thai workers had significantly higher levels than Chilean workers (adjusted—6.04 µg/g creatinine, 95% CI: 3.54, 8.54; log—1.21 µg/g creatinine, 95% CI: 0.81, 1.60). Independent t-tests between log levels of Pb in both blood and urine were used to compare participants with and without video recordings and indicated no statistically significant differences in Thailand or Chile. Thus, inferences from the analysis on the subsetted video data appear applicable to the total sample. Figure 2 visualizes the lead concentrations between video and non-video participants in Thailand and Chile.

Of the demographic variables among workers for whom video recordings were available, the typical number of hours worked in a week was significantly higher among Chilean workers than Thai workers by an average of 12.9 h (95% CI: 4.3, 21.5) (Table 3). On the other hand, the residence time was an average of 25.2 years higher among Thai workers than Chilean workers (95% CI: 15.0, 35.3). The frequency of those employed by a business was significantly higher in Chilean workers than Thai workers (χ^2^ = 5.9, df = 1, *p* < 0.05). Of the job characteristics evaluated from the video data, the use of a blunt striking instrument, T-wrench, pliers/scissors, and a power drill all occurred significantly more frequently among Thai workers than Chilean workers (*p* < 0.05 for all comparisons) (Table 4). Similarly, bending of the back, repetitive arm motion, squatting/kneeling, and sitting low to the ground were all performed significantly more frequently among Thai than Chilean workers (*p* < 0.05 for all comparisons). Thai workers worked near broken glass significantly more frequently than Chilean workers (0.5; 95% CI: 0, 1). Thai workers wore cotton gloves significantly more frequently while working, while Chilean workers wore close-toed shoes significantly more frequently (*p* < 0.01 for both comparisons). Finally, Thai workers dismantled random parts and electric fans significantly more frequently than the Chilean workers (*p* < 0.05 for both comparisons).

### 3.3. Boosted Regression Trees

Of the demographic and job characteristics predictors, twelve were highly correlated with each other (i.e., r > 0.70). These include handling sharp metal and removing sharp metal from electronics; printed circuit board and soldering iron; repetitive hand motion and repetitive arm motion; repetitive hand motion and constant hand grip; constant hand grip and repetitive arm motion; desktop monitor/TV flat screen and soldering iron; collecting broken glass and working near broken glass; desktop monitor/TV flatscreen and computer tower; desktop monitor/TV flatscreen and printed circuit board; noisy activities and blunt striking instrument; noisy activities and parts; and long pants and long sleeves. As a result, removing sharp metal from electronics, constant hand grip, long sleeves, desktop monitor/TV flatscreen, soldering iron, collecting broken glass, and noisy activities were dropped from the BRT models, leaving 45 predictors for analysis. The resulting randomness threshold was 100/45, or 2.22%.

Of the BRT model run on log Pb levels in urine, eight of the 45 predictors demonstrated variable importance above the 2.22% randomness threshold (Figure 3A). Partial dependence plots in Figure 3B(a–e) show the relationship of each of the five important job characteristics predictors from video analysis, controlling for all other predictors (where every other predictor is taken at mean value). The use of close-toed shoes contributed considerably to the model, with a relative importance of 43.9%, demonstrating a negative sigmoidal relationship with the log levels of Pb in urine (Figure 3B(a)). The use of a blunt striking instrument was the second most important variable, with a relative importance of 14% and a strongly positive, nearly linear association (Figure 3B(b)). The bending of the back while working showed a relative importance of 5.7% as well, with a positive, sigmoidal relationship to the log levels of Pb in urine (Figure 3B(c)). The dismantling of random parts in the workplace demonstrated a relative importance of 4.4%, with a concave-up parabolic association. The highest exposed workers were those who nearly always dismantled parts (Figure 3B(d)). Finally, bending the back at work displayed a relative importance of 3.5%, with a negative and almost linear association (Figure 3B(e)). The remaining partial dependence plots of the important demographic predictors are presented in the Appendix A. Briefly, the predicted log levels of Pb in urine were higher in Thailand than in Chile (Appendix A); residence time, in years, demonstrated a strong, positive association (Appendix A), and workers’ age showed a general decrease from the youngest workers to the oldest workers (Appendix A). Significantly, however, the model predicted an extreme spike in log levels of Pb in urine between the ages of 40 and 50. Twenty-two of the 45 variables had 0% relative importance. 

Of the BRT models run on log levels of Pb in blood, 15 of the 45 predictors demonstrated variable importance above the 2.22% randomness threshold (Figure 3C). The same five job characteristics shown in Figure 3B for the urine model are shown in Figure 3D(a–e). The bending of the neck demonstrated a relative importance of 9.6%, with a stagnant, non-linear association (Figure 3D(a)). Those who sometimes bend their necks were associated with the lowest log levels of Pb in blood. The dismantling of random parts was observed to have a relative importance of 8.9%, with a decreasing, concave-down association (Figure 3D(b)). The use of a blunt striking instrument demonstrated a relative importance of 5.6%, with a strong increasing association (Figure 3D(c)). The bending of the back while working demonstrated a relative importance of 5.3%, depicting an increasing, linear association (Figure 3D(d)). Wearing close toed shoes while working showed a relative importance of 5.1%, with a decreasing, linear association (Figure 3D(e)). The remaining ten partial dependence plots are presented in Appendix A. Briefly, some crucial findings from Appendix A are discussed. Residence time, in years, primarily contributed to the model with a relative importance of 15% and a strongly positive and linear relationship to log levels of Pb in blood (Appendix A). Those aged 50 and older were associated with the highest log levels of Pb in blood (Appendix A). Those working between 40 and 55 h per week were associated with the lowest log levels of Pb in their blood (Appendix A). Lastly, the use of screwdrivers and pliers/scissors both had a strong negative association with the log levels of Pb in blood (Appendix A, respectively). The same 22 variables from the urine model with 0% relative importance similarly had 0% relative importance in the blood model.

## 4. Discussion

### 4.1. Summary

This study sought to examine which characteristics of e-waste recycling workers and work activities are strongly associated with Pb levels in workers’ urine and blood. Overall, 16 of the 45 characteristics examined were important predictors of Pb in urine and blood. Among the eight characteristics important to Pb in urine, three were demographic (country, residence time, and age), one was mechanical (use of blunt striking instrument), two were musculoskeletal (bending the back and neck), one was related to PPE use (wearing close-toed shoes), and one was material- and activity-related (dismantling of random parts). Among the fifteen characteristics important to Pb in blood, five were demographic, three were related to tool use (the use of blunt striking instrument, screwdrivers, and pliers/scissors), five were musculoskeletal (bending of neck or back, lifting less or greater than 20 pounds, and repetitive arm motion), one was related to PPE use (close-toed shoes), and one was activity- and material-specific (dismantling of random parts). The use of close-toed shoes, blunt striking instruments, residence time, the bending of the back, age, the dismantling of random parts, and the bending of the neck were the seven of the 16 total important predictors overlapping both urine and blood models, suggesting that these factors may be the most influential factors on which to focus intervention efforts. The predictors related to chemicals or the potential for laceration were not found to be important. However, injuries are quite prevalent among e-waste workers, and may also be associated with higher toxic metal concentrations [9,16,32].

Overall, the Thai e-waste population had higher blood and urine Pb levels than the Chilean population. Compared to blood Pb levels among e-waste workers in Agbogbloshie, Ghana, the Thai e-waste population had slightly higher average blood Pb levels. In contrast, the Chilean population had lower average Pb levels [33]. However, compared to other studies in Agbogbloshie, both the Thai and Chilean e-waste populations had lower average blood Pb levels [34,35]. Compared to studies of e-waste workers in China, average blood Pb levels in the Thai and Chilean populations are also lower [35]. Other studies have also measured urine Pb levels among e-waste populations, but the measurements are not comparable to our study due to the different methodology in measuring Pb in urine or only measured Pb in blood [34,35].

While this appears to be the first e-waste study to have used video-based analysis to assess the relationship between work tasks and urine and blood lead concentrations, the video-based analysis has been utilized in many studies, particularly those regarding musculoskeletal disorders [36,37,38]. A 2016 study argues that the use of video surveillance to assess variables related to occupational safety offers the benefit of reducing the observer effect compared to in-person monitoring, and was found to be a valuable tool for identifying hazards [39]. Previous studies have also used video exposure assessment to explore the relationship between chemical exposures and workplace activities [40,41].

### 4.2. Implications

The levels of Pb in urine and blood were strongly influenced by wearing close-toed shoes. Among both models, the more frequently that close-toed shoes were worn while working, the lower the Pb levels in both urine and blood. Thai workers had higher Pb levels in their urine and blood than the Chilean workers while simultaneously wearing close-toed shoes significantly less frequently. Due to the rudimentary methods used in informal e-waste recycling in both countries, bits and pieces of electronics and their dust may quickly settle on the ground around workers. Ensuring that close-toed shoes are worn may reduce this route of exposure and lower levels of Pb among recycling workers.

The use of a blunt striking instrument to dismantle electronics, on the other hand, contributed to higher levels of Pb in urine and blood. Using a blunt striking instrument may generate Pb-containing aerosols and dusts, increasing the potential of Pb exposure by workers. These findings are in line with results from a study analyzing trace metals in a formal e-waste recycling shop in Hong Kong, which examined the use of blunt striking instruments on metallic dust suspended in the air and deposited on the floor. That study found that closer worker proximity to the ground was associated with higher levels of metal concentrations [14]. Congruent to these findings, more frequent use of non-striking tools such as screwdrivers and pliers/scissors during the dismantling of electronics contributed to lower Pb levels in blood. Thus, reducing the use of blunt striking instruments and dermal exposure to the ground through PPE use may be highly valuable to e-waste recyclers.

In both the urine and blood models, more frequent bending of the back led to higher levels of Pb. Bending the back can bring the part closer to the electronic components being disassembled and, therefore, to the Pb with which they would interact. Conversely, more frequent bending of the neck was negatively associated with Pb levels in urine, suggesting the possibility that workers with straight necks were further removed from Pb exposures than those with bent necks. No association was observed between the bending of the neck and blood Pb levels. 

The dismantling of random parts was the final work characteristic that importantly contributed to Pb levels in urine and blood. However, among urine, more frequently dismantling random parts was positively associated with Pb levels, while the inverse was true for blood measurements. Random parts were classified in the video process as anything that could not be clearly observed as a specific electronic; therefore, these random parts represent a gap in our understanding of e-waste recycling, and warrant further exploration.

Among the demographic variables, Pb levels in urine and blood were significantly influenced by the residence time and age of the e-waste worker. In both urine and blood, the longer the residence time of the worker, typically higher levels of Pb were predicted. While age was not strongly predictive in the positive or negative direction in urine and blood, ages around 40 to 50 had the highest Pb levels in both urine and blood. This population of workers may be the primary e-waste recyclers in their household who interact with e-waste more frequently and consistently than others. Interestingly, those who regularly performed e-waste recycling, approximately 40 to 55 h per week, were among those with the lowest predicted blood levels of Pb, while the opposite was true for those less than 40 h and above 55 h per week. Those who work less than a typical work week may be less experienced or more informal in their e-waste recycling methods, increasing the likelihood of higher exposure to Pb. In contrast, those who work more than a typical work week may be more exposed to Pb than their counterparts.

### 4.3. Limitations

There are four main limitations to this study. First, the sample size was relatively small (N = 225 workers split between Thailand and Chile). It may limit the overall generalizability of our results. Due to the logistical complexity and intensive effort required to collect and analyze work-shift videos, video recordings were available for only a subset of the sampled population (n = 40 workers). The small sample size also prevented us from performing any type of validation testing to examine how the models perform on external data. Second, the analysis was performed on video observations of job characteristics by five research assistants. While an acceptable level of inter-rater agreement was achieved, the subjectivity of this analysis must be recognized, along with the potential limitations of observing diverse work tasks from a video recorded at a single location over the course of a work shift. Future efforts may benefit from performing a quantification of work activities directly in the field rather than relying on recordings; although, a balance between in-person observations and video observations may help offset the limitations of each type of observation. Third, inferences from boosted regression tree modeling are associative in nature, rather than causal. Lastly, this study did not utilize a control group in neither Chile nor Thailand to compare Pb levels of the e-waste workers to the local, general public for each respective country. Future research of these sites would benefit from sampling background levels. Despite the limitations of these methods, the results are still valuable for identifying sub-populations within the informal e-waste recycling sector exposed to higher Pb levels and, therefore, in greater need of occupational health interventions.

## 5. Conclusions

Overall, our findings indicate five job characteristics that strongly influence Pb levels in e-waste recycling workers’ urine and blood, and five other job characteristics that are potentially strongly influential for blood levels of Pb. Workers’ ages, residence time, sex, the number of hours worked, and their socioeconomic status are also important factors to consider. Interventions which target these factors may help reduce the levels of Pb among informal e-waste recycling workers. Other considerations for future research include assessing toxic metals other than Pb, and considering multiple metals exposures.

## Figures and Tables

**Figure 1 ijerph-18-10580-f001:**
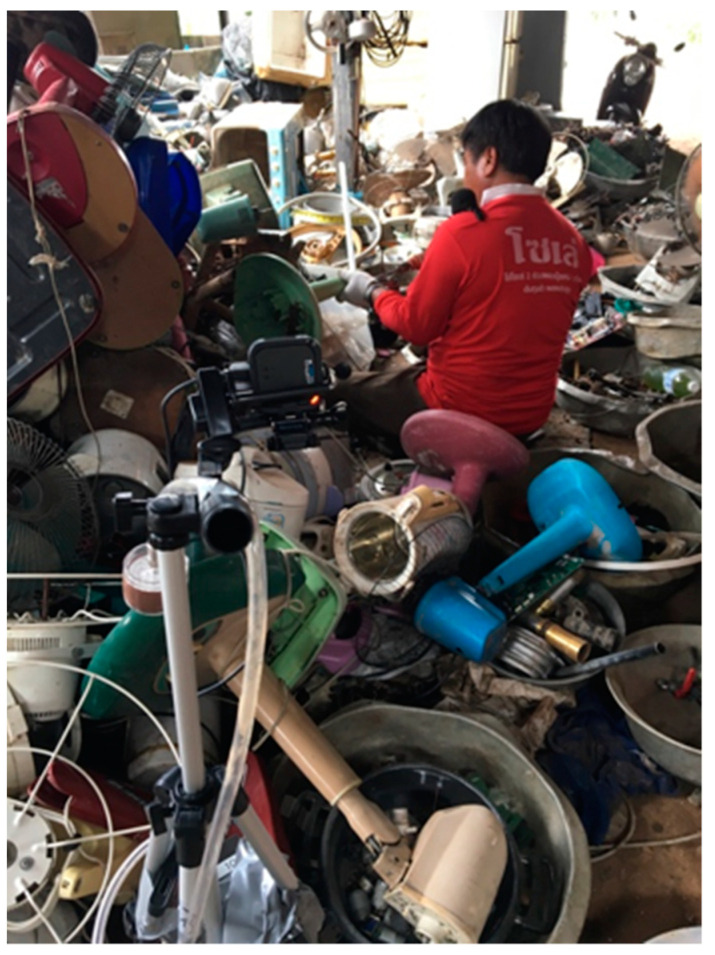
Video snapshot from a GoPro camera of an electronic waste (e-waste) worker dismantling e-waste during a work shift.

**Figure 2 ijerph-18-10580-f002:**
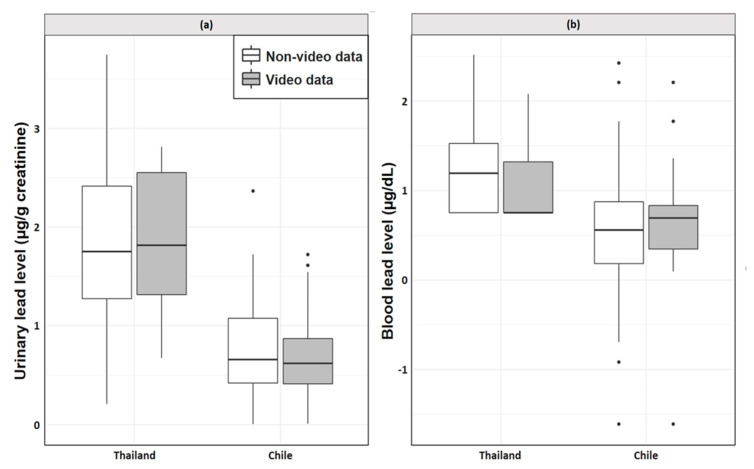
Boxplots of log lead concentrations in (**a**) urine (µg/g creatinine) and (**b**) blood (µg/dL) among video and non-video electronic waste participants in both Thailand and Chile.

**Figure 3 ijerph-18-10580-f003:**
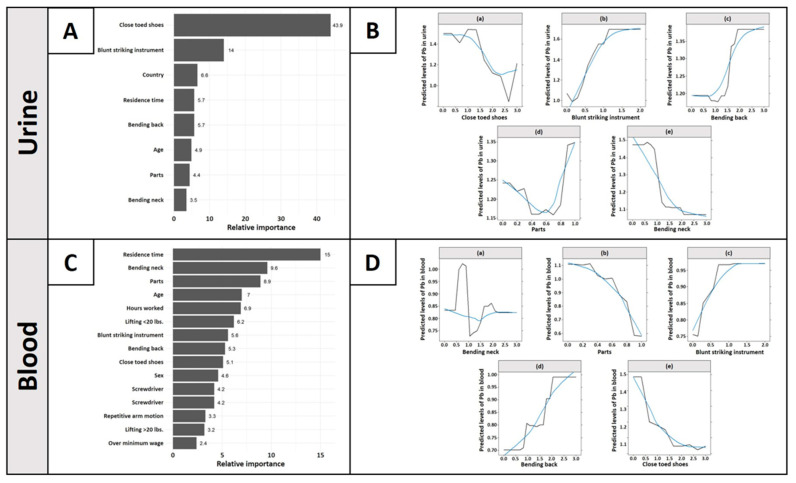
(**A**) The relative importance of demographic and job characteristics strongly predictive of log lead (Pb) concentrations in urine. The displayed factors exceeded the randomness threshold of 100/45, or 2.22%. (**B**) Partial dependence plots for the five more influential job factors for log levels of Pb in urine in order of decreasing relative importance: (a) use of close toed shoes; (b) use of a blunt striking instrument; (c) the bending of the back; (d) dismantling of random parts; (e) the bending of the neck. The black line signifies the unsmoothed partial dependence plot, while the blue line signifies the smoothed partial dependence plot. (**C**) The relative importance of demographic and job factors strongly predictive of log Pb concentrations in blood. (**D**) Partial dependence plots for five of the influential job factors for log levels of Pb in blood in order of decreasing relative importance: (a) the bending of the neck; (b) the dismantling of random parts; (c) the use of a blunt striking instrument; (d) the bending of the back; (e) use of close toed shoes.

**Table 1 ijerph-18-10580-t001:** Demographic characteristics of Thai and Chilean e-waste recycling workers.

Variables	Thailand			Chile		
	N	Mean (sd)	Median (Max)	N	Mean (sd)	Median (Max)
Body mass index (BMI; kg/m^2^) ^‡^	69	24.9 (4.2)	24.2 (39.0)	93	30.1 (5.4)	29.3 (57.6)
Residence time (years) ^‡^	123	41.7 (15.6)	43 (84)	93	14.4 (15.1)	8.0 (63.5)
Time in E-waste (years)	-	-	-	92	12.4 (11.81)	10.00 (42.0)
Age (years)	129	46.4 (12.5)	46 (84)	93	46.8 (14.3)	46 (87)
	**N**	**n (%)**		**N**	**n (%)**	
Sex (male) ^†^	130	71 (54.6%)		93	68 (73.1%)	
Marital-Single ^‡^	130	9 (6.9%)		87	19 (21.8%)	
- Married		115 (88.5%)			58 (66.7%)	
- Divorced		2 (1.5%)			1 (1.1%)	
- Cohabitating		0 (0%)			3 (3.4%)	
- Widowing		4 (3.1%)			3 (3.4%)	
- Separated		0 (0%)			3 (3.4%)	
Education-None	127	7 (5.5%)		93	7 (7.4%)	
- Primary		57 (44.9%)			30 (32.3%)	
- Secondary		48 (37.8%)			41 (44.1%)	
- Some college		15 (11.8%)			15 (16.1%)	
Employed by business (Yes) *	130	90 (69.2%)		93	90 (96.8%)	
Income > Min. Wage *	129	49 (38%)		87	55 (63.2%)	
E-waste job (Primary)	130	72 (55.4%)		50	34 (68.0%)	

Independent *t*-tests (for continuous variables) and χ^2^-tests (for categorical variables) performed between populations at * ~ *p* < 0.05, ^†^ ~ *p* < 0.01, ^‡^ ~ *p* < 0.001.

**Table 2 ijerph-18-10580-t002:** Urine and blood lead concentrations among Thai and Chilean e-waste recycling workers across all (full data) and video participants (video data).

Data	Sample Type		Thailand			Chile	
		N	Mean (sd)	Median (Max)	N	Mean (sd)	Median (Max)
Full data	Blood lead levels (µg/dL) ^†^	105	3.79 (1.99)	3.3 (12.4)	82	2.17 (1.71)	1.75 (11.3)
	Log-transformed ^†^	105	1.22 (0.46)	1.2 (2.5)	82	0.54 (0.71)	0.56 (2.42)
	Urine lead levels (µg/g creatinine) ^†^	105	7.45 (7.36)	4.77 (41.3)	86	1.41 (1.48)	0.93 (9.67)
	Log-transformed ^†^	105	1.83 (0.77)	1.75 (3.75)	86	0.74 (0.51)	0.66 (2.37)
Video data	Blood lead levels (µg/dL)	19	3.11 (1.54)	2.12 (8)	21	2.44 (1.92)	2 (9.1)
	Log-transformed *	19	1.05 (0.4)	0.75 (2.08)	21	0.66 (0.73)	0.69 (2.21)
	Urine lead levels (µg/g creatinine) ^†^	19	7.37 (5.07)	5.15 (15.67)	21	1.34 (1.35)	0.86 (4.6)
	Log-transformed ^†^	19	1.92 (0.69)	1.82 (2.81)	21	0.72 (0.51)	0.62 (1.72)

Independent *t*-tests performed between populations * ~ *p* < 0.05 and ^†^ ~ *p* < 0.001.

**Table 3 ijerph-18-10580-t003:** Demographics among Thai and Chilean e-waste recycling workers with video recordings.

Variables	Thailand (N = 19)			Chile (N = 21)		
	N	Mean (sd)	Median (Max)	N	Mean (sd)	Median (Max)
Age		45.3 (7.9)	46 (60)		50.8 (11.1)	51 (67)
Hours worked ^†^		40.7 (13)	40 (56)		53.6 (13.8)	50 (86)
Residence Time ^‡^	18	37.9 (15.2)	42 (60)		12.8 (16)	7.3 (63.5)
		**n (%)**			**n (%)**	
Education	18					
None		3 (15.8)			1 (4.8)	
Primary		5 (26.3)			7 (33.3)	
Secondary		7 (36.8)			8 (38.1)	
Some college		3 (15.8)			5 (23.8)	
Employed by business (Yes) *		12 (63.2)			21 (100)	
Income > minimum wage		7 (36.8)			14 (66.7)	
Sex (Male)		9 (47.7)			17 (81)	

Independent *t*-tests (for continuous variables) and χ^2^-tests (for categorical variables) performed between populations at * ~ *p* < 0.05, ^†^ ~ *p* < 0.01, ^‡^ ~ *p* < 0.001.

**Table 4 ijerph-18-10580-t004:** Job characteristics of Thai and Chilean e-waste recycling workers with video recordings.

Variables	Thailand (N = 19)			Chile (N = 21)		
	N	Mean (sd)	Median (Max)	N	Mean (sd)	Median (Max)
* Mechanical * ^§^						
Sharp Blade		0.06 (0.12)	0 (0.4)		0.01 (0.05)	0 (0.2)
Blunt striking instrument^‡^		0.88 (0.55)	1 (2)		0.13 (0.44)	0 (2)
T-wrench ^*^		0.09 (0.16)	0 (0.5)		0 (0)	0 (0)
Wrench		0.03 (0.05)	0 (0.1)		0.02 (0.06)	0 (0.2)
Pliers/scissors ^†^		0.32 (0.29)	0.3 (0.8)		0.07 (0.13)	0 (0.4)
Bolt Cutters		0.01 (0.05)	0 (0.2)		0 (0)	0 (0)
Chisel		0.19 (0.34)	0 (1)		0.06 (0.27)	0 (1.25)
Screwdriver		0.17 (0.22)	0.1 (0.8)		0.23 (0.31)	0.2 (0.12)
Power drill ^*^		0.23 (0.33)	0 (0.8)		0.02 (0.11)	0 (0.5)
* Musculoskeletal * ^§^						
Bending back ^‡^		1.89 (0.61)	1.9 (3)		0.93 (0.71)	0.9 (2.4)
Bending neck		1.5 (1.05)	1.8 (3)		1.06 (1)	0.8 (3)
Lifting < 20 lbs.		0.41 (0.59)	0 (2)		0.44 (0.59)	0.3 (2.58)
Lifting > 20 lbs.		0.59 (0.69)	0 (2)		0.42 (0.58)	0.2 (2.25)
Pushing/pulling		0.03 (0.08)	0 (3)		0.11 (0.2)	0 (0.8)
Repetitive arm motion ^†^		0.50 (0.64)	0.2 (2)		0.05 (0.1)	0 (0.4)
Repetitive hand motion		0.56 (0.79)	0.2 (2.4)		0.19 (0.38)	0 (1.6)
Squatting/kneeling ^*^		0.82 (1.22)	0.1 (3)		0.12 (0.26)	0 (0.9)
Sitting low to ground ^‡^		1.79 (1.34)	2.6 (3)		0.01 (0.05)	0 (0.2)
* Chemicals * ^§^						
Use of Chemical		0.01 (0.03)	0 (0.1)		0 (0.02)	0 (0.1)
* Potential for Lacerations * ^§^						
Working near broken glass ^*^		0.52 (1.04)	0 (3)		0.01 (0.07)	0 (0.3)
Working near sharp metal		0.81 (1.33)	0 (3)		0.47 (0.81)	0 (2.4)
Handling sharp metal		0.15 (0.35)	0 (1)		0.04 (0.1)	0 (0.3)
* PPE * ^§^						
Cotton gloves ^‡^		2 (1.4)	2.9 (3)		0.33 (0.79)	0 (3)
Close toed shoes ^†^	18	1.37 (1.46)	0.6 (3)	20	2.52 (0.91)	3 (3)
Rubber Gloves		0.44 (1.05)	0 (3)	20	0.3 (0.79)	0 (3)
Fabric as mask		0.27 (0.81)	0 (2.7)		0 (0)	0 (0)
Dusk mask		0.16 (0.69)	0 (3)		0 (0)	0 (0)
Long pants		2.31 (1.23)	3 (3)		2.64 (0.66)	3 (3)
* Electronics * ^||^						
TV - CRT		0 (0)	0 (0)		0.04 (0.15)	0 (0.67)
Washing machine		0.05 (0.23)	0 (1)		0.02 (0.05)	0 (0.2)
Electric Fan ^*^		0.29 (0.45)	0 (1)		0 (0)	0 (0)
Desktop Monitor - CRT		0.01 (0.02)	0 (0.1)		0.01 (0.03)	0 (0.1)
Computer Tower		0.01 (0.03)	0 (0.1)		0.01 (0.04)	0 (0.2)
Cellphone		0 (0)	0 (0)		0.05 (0.22)	0 (1)
Laptop		0 (0)	0 (0)		0.04 (0.17)	0 (0.8)
Printed circuit board		0 (0)	0 (0)		0.06 (0.15)	0 (0.6)
Parts ^†^		0.81 (0.36)	1 (1)		(0.37)	0.3 (1)

Independent *t*-tests (for continuous variables) and χ^2^-tests (for categorical variables) performed between populations at ^*^ ~ *p* < 0.05, ^†^ ~ *p* < 0.01, ^‡^ ~ *p* < 0.001, ^§^ Frequency scale ranges from 0–3, ^||^ Frequency scale ranges from 0–1.

## Data Availability

The data supporting reported results can be made available to reasonable requests by emailing the corresponding author of this paper, Aubrey Arain, at arain@oakland.edu.

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
