# Peer review of "Work Task Association with Lead Urine and Blood Concentrations in Informal Electronic Waste Recyclers in Thailand and Chile"

_ijerph, 2021, doi:10.3390/ijerph182010580_

Round 1

Reviewer 1 Report

  1. In the Thai and Chilean e-waste recycling workers, the test results on the lead concentration in urine and blood in the general public, a control group, are missing. Can you add data on this? Because it is thought that data on how high the waste e-waste recycler appears to the general public by handling the waste should be preceded.

      2. Boosted regression trees (BRT), which was suggested as a method of machine learning, was used. I think that the rationale for how reliable this method is should be provided. Please provide the program name and additional references for using BRT.

Reviewer 2 Report

The manuscript entitled ”Work task association with lead urine and blood biomarker concentrations in informal electronic waste recycles in Thailand and Chile” written by Shkembi et al. is well-written paper that has significant value. However, some points must be improved.

1. The term “biomarker” refers to a subcategory of medical signs. It is an objective indication of medical state observed from outside the patient – which can be measured accurately and reproducibly. What authors had in mind by writing blood biomarker concentrations? You can check for ex ample hemoglobin or hematocrit to check if patient has anemia, or you can check creatine to check how patient kidney working. But Pb is not biomarker. It is a heavy metal, which does not play any physiological role in the human body. Please, delete this term from the manuscript.

2. The title, aim, results and discussion is focused only on Pb concentration. But in material and method section, you wrote that you examined the concentration of Al, Fe, Mn, Ni, Cu, and Zn. Please, correct it.

3. Line 368: "average Pb lead levels [29]” Delete either Pb or lead

Round 2

Reviewer 2 Report

Thank you for considering my comments.